# Smart Temperature Sensor Design and High-Density Water Temperature Monitoring in Estuarine and Coastal Areas

**DOI:** 10.3390/s23177659

**Published:** 2023-09-04

**Authors:** Bozhi Wang, Huayang Cai, Qi Jia, Huimin Pan, Bo Li, Linxi Fu

**Affiliations:** 1Institute of Estuarine and Coastal Research, School of Ocean Engineering and Technology, Sun Yat-Sen University, Guangzhou 510275, China; wangbzh5@mail2.sysu.edu.cn (B.W.); panhm5@mail2.sysu.edu.cn (H.P.); libo33@mail2.sysu.edu.cn (B.L.); fulx6@mail.sysu.edu.cn (L.F.); 2Guangdong Provincial Engineering Research Center of Coasts, Islands and Reefs, Guangzhou 510275, China; 3Southern Laboratory of Ocean Science and Engineering (Zhuhai), Zhuhai 519082, China; 4SiSensor Technology Company, Zhuhai 519082, China; jiaqi@sisensor.cn

**Keywords:** low-cost sensors, high-accuracy sensors, high-sensitivity sensors, long-term, high-density monitoring

## Abstract

Acquiring in situ water temperature data is an indispensable and important component for analyzing thermal dynamics in estuarine and coastal areas. However, the long-term and high-density monitoring of water temperature is costly and technically challenging. In this paper, we present the design, calibration, and application of the smart temperature sensor TS-V1, a low-power yet low-cost temperature sensor for monitoring the spatial–temporal variations of surface water temperatures and air temperatures in estuarine and coastal areas. The temperature output of the TS-V1 sensor was calibrated against the Fluke-1551A sensor developed in the United States and the CTD-Diver sensor developed in the Netherlands. The results show that the accuracy of the TS-V1 sensor is 0.08 °C, while sensitivity tests suggest that the TS-V1 sensor (comprising a titanium alloy shell with a thermal conductivity of 7.6 W/(m °C)) is approximately 0.31~0.54 s/°C slower than the CTD-Diver sensor (zirconia shell with thermal conductivity of 3 W/(m °C)) in measuring water temperatures but 6.92~10.12 s/°C faster than the CTD-Diver sensor in measuring air temperatures. In addition, the price of the proposed TS-V1 sensor is only approximately 1 and 0.3 times as much as the established commercial sensors, respectively. The TS-V1 sensor was used to collect surface water temperature and air temperature in the western part of the Pearl River Estuary from July 2022 to September 2022. These data wells captured water and air temperature changes, frequency distributions, and temperature characteristics. Our sensor is, thus, particularly useful for the study of thermal dynamics in estuarine and coastal areas.

## 1. Introduction

The water temperature is one of the most relevant and dominant factors that affect the physical, chemical, and biological processes in various types of water bodies (e.g., ocean, estuaries, rivers, and lakes) and is a clear indicator of climate change. Thus, it is essential to quantify the spatial–temporal dynamics of the water temperature and the relationship between the water temperature and aquatic ecosystems [1,2,3,4]. There exists a long tradition of monitoring using satellite water surface temperature measurements to understand the responses of aquatic communities (such as coral reefs and macroalgae communities) to thermal stress events as a direct result of global warming [5,6]. Remotely sensed satellite water surface temperature measurements are advantageous in quantifying the broadscale pattern of the water surface and its relationship to biological patterns globally; however, the application at small spatial scales (such as 10 × 10 m resolution) is subject to inaccuracies due to limited spatial resolution. In addition, satellites only measure the near-surface layer of the water body and may not reliably measure the potential impacts of the water temperature on benthic ecosystems. The combination of remote sensing with in situ methods can improve the quality of water temperature monitoring activities [7,8]. Previous studies have shown that water temperature records based on available in situ measurements are effective for many purposes provided there is sufficient instrument spacing and precision [9,10,11]. Water temperature monitoring can be improved (especially for the management of coastal protected areas) through the use of more precise measurements at a greater spatial density [12]. However, this comes at a greater cost, since the water temperature is commonly measured with resistance temperature detectors (RTDs), such as the Pt100 or Pt1000, or thermistors, which are generally precise but expensive for larger-scale and high-density applications [13,14,15,16,17]. It should be noted that there exist many different types of semiconductor sensors with various structures. For instance, P-N junction temperature sensors, achieved using materials such as 4H-silicon carbide or a SOI-CMOS (complementary metal oxide semiconductor), cover temperatures of 20 to 600 °C with 3.5 mV/°C sensitivity, albeit being susceptible to noise and drift [18,19,20]. Schottky diodes offer similar sensitivity across temperatures of −65 to 115 °C, with the accuracy being impacted by both pressure and humidity [21]. Fiber optic sensors can operate at up to 800 °C with sensitivity rates of 0.06–0.46 nm/°C, using materials such as fused silica, cones, or graphene for compactness, sensitivity, and corrosion resistance. However, the challenges include the susceptibility to optical fiber deformations and rather high costs [22,23]. We developed a low-power (with the main module’s power consumption being approximately 130 μA for a duration of roughly 10 ms), high-precision, yet low-cost smart temperature sensor using mainstream CMOS technology [24,25,26], which enables precise water temperature monitoring at a lower cost and greater spatial density.

In this study, we deployed 26 smart CMOS sensors in the estuarine and coastal areas of the Pearl River Delta, situated in the southern part of China. Both indoor controlled tests and in situ deployments were adopted to demonstrate the accuracy of the developed smart sensor, here named TS-V1. In addition, we also illustrate the utility of having access to big data collected by several low-cost sensors by presenting results that demonstrate the fine-scale variability of water temperatures measured in the Pearl River Estuary.

## 2. Materials and Methods

### 2.1. CMOS Technology and Operation of the TS-V1 Sensor

A high-quality circuit design is desirable for developing a low-power, low-cost, yet highly accurate smart sensor. Temperature sensors can be packaged similarly to other integrated circuits and readily constructed using standard CMOS technology. A wide range of intelligent temperature sensors are available as low-cost CMOS technologies. For more details, readers can refer to [27].

Briefly, we overview the general facts about integrated circuits and focus on the most crucial part for improving the circuit accuracy and reducing the power consumption. In particular, the voltage *U* (proportional to the absolute temperature, *V*) of the CMOS chip can be converted to temperature *T* (°C) using the following equation [24,28]:(1)$$\Delta U_{BE}=\frac{kT}{q}\ln p$$
where ∆*U*_BE_ is an intrinsically accurate function of the absolute temperature (*U*_BE_ is the base emitter voltage, μV), *k* is Boltzmann’s constant (J/K), *q* is the electron charge (C), and *p* is the emitter current ratio (i.e., *p* = *Q*_1_:*Q*_2_, *Q*_1/2_ are two identical bipolar transistor currents, mA). Subsequently, a temperature-independent band gap reference voltage (*U_REF_*) can be obtained by combining the *V_BE_* of another transistor *Q*_3_ with a scaled version of ∆*U_BE_*. Finally, an ADC (analog-to-digital converter) determines the ratio of *α**∆*V_BE_* and *V_REF_* to obtain a digital output *D_temp_* that is proportional to the temperature, as follows:(2)$$D_{\mathrm{temp}\;}=\frac{a\cdot \Delta U_{BE}}{U_{BE}+a\cdot \Delta U_{BE}}=\frac{U_{PTAT}}{U_{REF}}$$
where *U_PTAT_* is the correction voltage (μV) and *α* is the required scale factor established by appropriately sizing the sampling capacitors at the input of the modulator.

In CMOS chip circuits, the straightforward implantation of MOSFETs (metal oxide semiconductor field-effect transistors) results in poor accuracy. Fortunately, this fault can be solved by applying DEM (dynamic element matching), chopping, and trimming, among other approaches. (Figure 1) [20]. After a single temperature trim, the sensor’s accuracy can be as high as ±0.1 °C (with a range of −10 to 65 °C). These parameters are adjusted within each chip through a technology that mitigates circuit imperfections and offers universal applicability. This method relies on an array of sensors for testing. The amplifier, recurrent in the signal conditioning circuit, is used multiple times. Analog-to-digital conversion is integrated into the chip, producing a digital signal. Ultimately, the individual calibration of each chip is essential for enhanced factory accuracy and sensor precision. The specifications concerning the calibrated CMOS chip are detailed in Table 1.

### 2.2. The Structure of the Smart Temperature Sensor

Figure 2 shows the assembled module of the TS-V1 sensor. The integrated sensor consists of an MCU (microcontrol unit), an interface for connecting batteries and communication, sensor chips, and electric power, corresponding to the top board, middle board, bottom board, and battery, respectively (see Figure 2a,b). To be more specific, the TS-V1 sensor incorporates a temperature probe (depicted in Figure 2a ⑦) with the ability to directly perceive and record temperature variations. This probe establishes a physical connection with the data storage chip (shown in Figure 2a ④) via a cable. The internal architecture involves using a low-speed crystal oscillator (illustrated in Figure 2a ⑤) to provide highly accurate timing signals to the microcontroller (an inherent element within Figure 2a ①). Meanwhile, a high-speed clock crystal (depicted in Figure 2a ③) supplies timing signals to the integrated internal counter. In the operational mode, the low-power Cortex M0 microcomputer deals with the acquired temperature data. In addition, the system includes a low-power linear voltage regulator chip (depicted in Figure 2a ⑥) that converts the battery voltage into a stable output voltage. Such a stable voltage is distributed to each component within the whole system, ensuring consistent power supply throughout the system.

The TS-V1 is relatively easy to build with basic soldering knowledge. The first step is to connect the top board and the middle board by using five pin headers. Next, the top board is assembled to the bottom board using three copper wires, keeping space between the middle board and the bottom board for the battery. Two copper wires are then cut to the desired length and used for the tip and ring, respectively. Here, the tip and ring are attached to the battery’s positive terminal at one end and connected to the middle board at the other end. Finally, the circuit boards and battery are installed inside a titanium alloy cable using silicone. The silicone used is high-temperature-resistant, insulating, and waterproof. The silicone cures after two days, after which a pH test strip is attached to the inside of the titanium cap (see Figure 2c). 

### 2.3. Experimental Design of Measurement Parameters and Deployment of the TS-V1 Sensor

We assessed and compared the accuracy, sensitivity, and stability of the TS-V1 and CTD-Diver sensors [29].

The accuracy experiments were carried out in a controlled manner characterized by temperature fluctuations and substantial variations in temperature. The TS-V1 temperature sensor serves the purpose of measuring ocean surface temperatures primarily within temperate to subtropical regions. Given that the sea temperatures in these areas rarely reach freezing conditions, the temperature range of −10 to 60 °C adequately addresses the requirements of our intended applications. A detailed description of these experimental conditions is provided in Figure 3a. We conducted temperature tests using a temperature control box, wherein the temperature was gradually increased from −10 to 60 °C with 10 °C increments. Each temperature was maintained for a duration of 5 h.

The sensitivity experiments for the temperature measurements involved heating and cooling tests, and included separate systems for testing the water temperature and air temperature using a thermostat. In the water temperature test, two uncovered tanks were filled with water at temperatures of 30 °C and 50 °C, respectively (Figure 3b). The sensors were subjected to a sequential series of temperature exposures, starting at 30 °C, transitioning to 50 °C, and returning to 30 °C, with each phase lasting for a duration of 30 min. This whole process was repeated twice. The air temperature test followed a similar procedure as the water temperature test but it was performed within a hollow aluminum box within the thermostat (Figure 3a), with each phase lasting for a duration of 30 min. 

To ensure precision, we adopt the measured temperature from the Fluke-1551A thermometer (see the link in [30]), which offers an accuracy rate of ±0.05 °C, as the reference. The Fluke-1551A thermometer supplied real-time incubator temperature readings and was allowed to stabilize before recording. This procedure facilitates a reliable comparison and calibration process.

In order to assess the stability of the temperature sensors, a total of 26 devices were installed in the downstream basin of the Modaomen Estuary (MDM), Jitimen Estuary (JTM), and Hutiaomen Estuary (HTM), situated in the south-western part of the Pearl River Delta, China (see Figure 4). The installation period was from 12 July 2022, to 25 September 2022. A significant number of sensors were deployed to achieve a high-density spatial configuration, ensuring thorough coverage across the area of interest. Each site was equipped with two sensors, with one sensor positioned approximately 2 m below the water surface and another sensor positioned 2 m above the water surface. The deployment depths for the underwater sensors at each site were determined to be below the lowest water level during spring tides. The distance between the two sites was approximately 8 km. Furthermore, two CTD-Divers were installed at M5 for comparative analysis with the TS-V1 sensor.

### 2.4. Generalized Extreme Value (GEV) Distribution

The extreme value theory was first proposed by Fisher and Tippett [31] and was further developed by Gnedenko [32]. It is based on the Fisher–Tippett–Gnedenko theorem, which says that if *X*_1_, *X*_2_, …, *X_n_* are samples of independent and identically distributed random variables and *M_n_* = max {*X*_1_, *X*_2_, …, *X_n_*} denotes the maximum value among *n* sampled observations, then the distribution of *M_n_* converges to the GEV distribution:(3)$$G(x)=\left\{\begin{array}{cc} \exp\left\{-{\left[1+g\left(\frac{x-m}{s}\right)\right]}^{-1/g}\right\} & \;\mathrm{if}\;g\neq 0\\ \exp\left[-\exp\left(-\frac{x-m}{s}\right)\right] & \;\mathrm{if}\;g=0 \end{array}\right.$$
where *m* and *s* denote location and scale parameters, respectively, while *g* is a shape parameter. The location parameter *m* represents the mean of the fitted distribution, indicating the central tendency of the observations. The scale parameter *s* controls the data variability and influences the steepness of the curve. Higher *s* values result in a broader distribution. The shape parameter *g* indicates the type of distribution. For example, *g* = 0 signifies a Gumbel distribution, *g* > 0 suggests a Frechet distribution, and *g* < 0 indicates a Weibull distribution. In addition, *g* determines the shape of the distribution’s tail. Higher *g* values lead to heavier tails, with more probability mass in the extreme values, while lower *g* values result in lighter tails with less probability mass in the extremes [33].

## 3. Results and Discussion

### 3.1. Accuracy, Sensitivity, and Stability of the Smart Temperature Sensors

The accuracy of the TS-V1 and CTD-Diver sensors can be quantified by determining the discrepancies between the reference values observed by the Fluke-1551A sensor and the measurements recorded by the other two devices. The accuracy rates of the TS-V1 and CTD-Diver instruments under various temperature scenarios are displayed in Figure 5. The degree of accuracy is higher when the sensors operate under temperatures of 0 or 60 °C. In contrast, the errors are relatively larger for temperature scenarios at 20 or 40 °C. Overall, it can be inferred that the accuracy of the TS-V1 device is consistently larger than the CTD-Diver device throughout the duration of operation. As the temperature rises above 0 °C, the error rates of both devices follow a U-shaped distribution, with their highest degrees of accuracy occurring at lower or higher temperatures. Within the temperature control range of −10 to 60 °C, the TS-V1 exhibits a maximum temperature shift error of approximately ±0.03 °C when compared to the Fluke-1551A reference standard (the Fluke-1551A sensor itself maintains an accuracy rate of approximately ± 0.05 °C across its entire temperature range). In comparison, the CTD-Diver device demonstrates a maximum error of ±0.05 °C in temperature measurements relative to the Fluke-1551A sensor.

The averaged deviation between the TS-V1 device and the Fluke-1551A reference values across a temperature range of −10 to 60 °C is approximately −0.012 °C, whereas for the CTD-Diver device, it is approximately 0.022 °C. These results suggest that the temperature accuracy of the TS-V1 sensor is acceptable, and it can be used for precise temperature monitoring purposes.

Figure 6 presents the TS-V1 and CTD-Diver sensors’ responses to water and air temperature changes, as assessed using heating and cooling experiments. In these tests, smaller values indicate a shorter time required for a 1 °C temperature change, indicating faster adaptation to temperature fluctuations. As the water temperature increases, the sensitivity of TS-V1 is approximately 0.54 s/°C slower when compared to CTD-Diver during heating tests, as depicted in Figure 6a. When the water temperature decreases, the CTD-Diver device responds 0.31 s/°C faster than the TS-V1 sensor, which is similar to the heating results. For air temperatures, the sensitivity rates of TS-V1 are 10.12 s/°C and 6.92 s/°C quicker than those of CTD-Diver in the cold and heating assessments, respectively, as depicted in Figure 6b. The difference might be attributed to the distinct shell compositions of the TS-V1 and CTD-Diver sensors. It is imperative to underscore that the presence of a titanium alloy shell engenders no deleterious ramifications regarding the chip’s performance; rather, it proffers a safeguarding influence. Our chip employs iterative low-fropout (LDO) and CMOS processing protocols. This dual-pronged approach achieves two-fold objectives: firstly, it guarantees an exceedingly low voltage magnitude across the chip’s internal circuitry, thereby virtually mitigating any self-induced thermal escalation; secondly, the LDO treatment ensures that alterations to the external circuit heat or voltage hold negligible sway over the internal chip circuitry. 

Regarding the thermal attributes intrinsic to the shell material, in the context of water temperature measurement, it is apparent that the water’s specific heat capacity, measuring 4.3 kJ/(kg °C), markedly surpasses that of the titanium alloy (0.5 kJ/(kg °C)) and zirconia (0.8 kJ/(kg °C)). This empirical disparity substantiates the provision of ample thermal energy by the aqueous medium for sensor temperature measurements. Furthermore, the titanium alloy exhibits a swifter temperature response than zirconia under equivalent water temperature conditions. Regarding the thermal conductivity of the housing, the TS-V1 sensor’s titanium alloy shell has a thermal conductivity rate of 7.6 W/(m °C), whereas CTD-Diver’s zirconia shell has a value of 3 W/(m °C). Previous studies have suggested that materials possessing higher thermal conductivity have a greater propensity for heat absorption or dissipation [34]. 

Beyond the heat capacity and thermal conductivity, the titanium alloy shell confers additional fortification to the chip; the selection of the titanium alloy emanates from its robust structural endurance, thereby endowing resilience against heightened aquatic pressures and effectively shielding the internal temperature sensor. Notably, the post-curing phase of the silicone encapsulant imparts minimal stress and bestows no deleterious ramifications upon chip performance.

However, our experimental outcomes do not align with this expectation, as they do not demonstrate that the TS-V1 sensor has a faster response rate compared to the CTD-Diver sensor. The CTD-Diver probe’s architectural design appears to influence its sensitivity. The CTD-Diver detector head temperature probe’s positioning diameter is more substantial compared to the other components, accompanied by a thinner casing (Figure 7). The aforementioned dual design features enhance the heat conduction surface area of CTD-Diver. In contrast, TS-V1’s shell employs a fully enclosed cylindrical structure, with no distinct design for the temperature probe, resulting in its reduced sensitivity to temperature increases compared to CTD-Diver. Figure 8a provides additional insights into this hypothesis. The response of the CTD-Diver and TS-V1 instruments during the water temperature cooling experiment exhibited variations across three distinct phases. In the initial and intermediate phases, CTD-Diver demonstrated a response similar to that of TS-V1 (Figure 8a). However, in the final stages of the heating or cooling experiment, CTD-Diver responded faster than TS-V1. This suggests that the hollow structure of the CTD-Diver sensor demonstrates enhanced sensitivity in detecting subtle temperature changes, particularly when there is a small temperature difference between the sensor and its surrounding environment. 

In the air temperature sensitivity test, CTD-Diver’s shell has a specific heat capacity of approximately 0.8 kJ/(kg °C), which is similar to that of air (which is 1.4 kJ/(kg °C)). This implies that air is not capable of consistently providing sufficient energy to facilitate the heating or cooling of the CTD-Diver sensor’s shell. This results in a slower response of CTD-Diver to changes in air temperature as compared to TS-V1 (with a heat capacity of approximately 0.5 kJ/(kg °C)).

Moreover, we have also shown that the sensitivity deviations among the CTD-Diver samples were approximately 0.08 °C (first heating test), 0.33 °C (first cooling test), 0 °C (second heating test), and 3.85 °C (second cooling test). On the other hand, for the TS-V1 samples, the sensitivity errors were approximately 0.08 °C, 2.74 °C, 0.08 °C, and 0.83 °C, respectively. Notably, the air temperature sensitivity experiment revealed a minimal difference (almost 0) in sensitivity between the two instruments.

After two temperature heating and cooling cycles, TS-V1 showed ±0.07 to ±0.1 hysteresis rates in terms of water temperature sensitivity, while the CTD-Diver sensor showed ±0.02 to ±0.09 hysteresis rates. For air temperature sensitivity, TS-V1 exhibited ±0.36 to ±0.4 hysteresis rates, while CTD-Diver showed ±0.25 to ±0.3 hysteresis rates.

The temperature readings acquired by TS-V1 were compared with those of the commercially available CTD-Diver sensor. We utilized the quantile-on-quantile (QQ) regression method to illustrate the stability of the TS-V1 sensor. In Figure 9, the plus (‘+’) symbols represent measured data points. They feature a solid black line that connects the first quartile (25th percentile) and the third quartile (75th percentile). This drawn line graphically portrays the ordering of the temperature data throughout the observation period, emphasizing the temperature spectrum that covers the 25th to 75th percentiles. Meanwhile, temperature data falling below the 25th percentile or exceeding the 75th percentile are displayed using a black dashed line. Generally, these drawn and dashed black lines visually display the theoretical temperature distribution of TS-V1, suggesting coherence between the observations derived from the CTD-Diver and TS-V1 devices. The QQ plot demonstrates that there is a strong correlation between the observations obtained by TS-V1 and CTD-Diver, with linear coefficients of determination r of 0.98 and 1.01 being observed for water and air temperatures, respectively (Figure 9).

Interestingly, the TS-V1 measurements are of higher magnitudes than the CTD-Diver measurements under low water temperatures and very high air temperatures. The TS-V1 sensor has a slight instability under low water temperatures (as shown in Figure 9a) and moderate to high air temperature conditions (as shown in Figure 9b). Of particular note is its higher measurement values relative to the CTD-Diver sensor under low water temperature conditions. Conversely, when the air temperature is at a moderate level, the TS-V1 measurements have a slightly lower magnitude than the CTD-Diver measurements. This suggests that under extreme temperature conditions, the TS-V1 data are more prone to fluctuations induced by variations in water or air temperature. However, the stability of the TS-V1 sensor should not be a cause for concern. When the water temperature range is 27–28 °C, CTD-Diver and TS-V1 show only minor differences, with an error range of 0.3 °C. However, at air temperatures of 41–42 °C, a notable 1.73 °C difference exists between the CTD-Diver and TS-V1 sensors. When the water temperatures are between 27 and 28 °C, the CTD-Diver and TS-V1 sensors show only minor differences, with an error range of 0.3 °C. However, at air temperatures of 41–42 °C, a notable 1.73 °C difference exists between the CTD-Diver and TS-V1 sensors. This discrepancy arises from their intrinsic semiconductor traits, causing temperature-dependent drift linked to their distinct designs [27,35,36].

Nevertheless, the outcomes of the accuracy experiments distinctly showed the heightened precision of TS-V1 when juxtaposed with CTD-Diver. Therefore, the discerned inadequacies in the alignment of the QQ plot between CTD-Diver and TS-V1 at the temperature juncture of 40 °C can be primarily attributed to the CTD-Diver sensor. As air temperature increases, the greater temperature sensitivity of TS-V1 to heat absorption leads to a more significant temperature rise than for CTD-Diver. The above errors may also be affected by the binding method. In order to bind the CTD-Diver and TS-V1 sensors on the same rope, we tied the TS-V1 sensor above the CTD-Diver sensor. This inconsistency may have been due to the specific heat capacity of TS-V1 being smaller than that of the CTD-Diver sensor.

It needs to be added that CTD-Diver requires a plastic protection device for the probe, which can detach easily and has a fragile ceramic shell. Fluke-1551A is suitable for indoor use with a 300 h battery life, although its probe is vulnerable to damage. Conversely, the design of TS-V1 optimizes the CMOS-based circuit chip structure, supplemented by the LDO configuration, enabling high accuracy, low power consumption, and minimal noise. Notably, the sensor displays limited data drift across a wide range of temperatures (with only considerable drift emerging above 100 °C), which allows for complex marine environment monitoring. The titanium alloy shell ensures efficient thermal conductivity and sustained sensitivity. Further bolstered by its cost-effectiveness and streamlined manufacturing process, the proposed TS-V1 sensor is well-suited for comprehensive temperature monitoring on a broader scale when compared with the commercial alternatives.

### 3.2. Long-Term and High-Density Monitoring of Water and Air Temperatures

Statistically significant differences were observed in the air and water temperature distributions among the sampling sites during the study period. The divergent temperature patterns between the water and air temperatures at these sites emphasize the importance of complex factors influencing estuarine and coastal water heat budgets, rather than assuming that water temperatures respond similarly to air temperatures alone. Table 1 displays the shape (*g*), location (*m*), and scale (*s*) parameters based on the GEV distribution. The *g* value of the GEV distribution of the water temperature is consistently less than 0, indicating a Weibull distribution. Most of the temperature observations follow a Weibull distribution. However, in specific areas such as the middle and upper reaches of the MDM, the upper reaches of the JTM, and the middle reaches of the HTM, a Frechet distribution can be observed. In addition, the distribution range (*s*) of the air temperatures at each station is wider than that of the water temperature (Table 2). Figure 10 illustrates the different ranges of water and air temperatures observed in the MDM estuary, JTM estuary, and HTM estuary. The range of water temperatures is notably narrower, with values ranging from 26 °C to 35 °C, compared to the air temperatures, ranging from 24 °C to 40 °C. 

A notable disparity was observed in the highest temperature frequencies for water and air during the study period. Figure 10 provides a comparison of the water temperature and air temperature distributions. The calibrated GEV distribution parameters are presented in Table 1. The water temperature distribution exhibits a higher concentration of observations in the high-temperature range compared to the air temperature distribution. This emphasizes the distinct thermal characteristics of the two systems, with the water temperatures consistently maintaining higher values while the air temperatures demonstrate more variability and a wider range of values. Specifically, the frequency peaks of the water temperatures occurred at 30~31 °C in the MDM, 29~30 °C in the JTM, and 30~31 °C in the HTM, respectively. The corresponding temperature distribution probability exhibited a significant skew towards higher values at the outlet location relative to the upper reaches. In most stations within the three sub-estuaries, the frequency distribution of the water temperatures confirmed the near-normal distribution, except for some stations (see Figure 10b,g,j,m), which exhibited a right-skewed distribution. The temperatures corresponding to the peak frequency of the air temperature measurements were observed at 28~30 °C in the MDM, 29 °C in the JTM, and 28~29 °C in the HTM, respectively. In addition, the peaks were skewed towards the middle or upper reaches, rather than the outlet location. The spatial distribution of the highest frequency of water and air temperatures indicates that factors other than the air temperature may considerably contribute to the changes in water temperature.

Figure 11 shows a comparison of the mean, range, and standard deviation values of temperatures across deployment sites (M1-M7) in the MDM, (J1–J3) JTM, and (H1–H3) HTM estuaries using the TS-V1 temperature sensor. The results show that the mean water temperature (30.35 °C) for each station closely approximates the corresponding air temperature (30.5 °C). The range curves of the water and air temperatures’ spatial variations in the MDM estuary show a similar pattern at the mouth, although the JTM and HTM estuaries demonstrate a significant contrast. Specifically, within the Xijiang Estuary’s channel networks, the JTM station displays the smallest (J2) and largest (J1) disparities between the water temperature and air temperature ranges. The minimum temperature difference recorded at this estuary was 5.96 °C, while the maximum anomaly was 17.8 °C. The MDM had the smallest standard deviation in air temperature, followed by the JTM with a slightly larger deviation, while the HTM had the largest deviation. Similarly, for water temperature, the JTM had the smallest deviation, the MDM had a slightly larger deviation, and the HTM exhibited the largest deviation. Furthermore, the range and standard deviation of the water temperatures were lower compared to the air temperatures, being 10.79 °C and 2.24 °C, respectively, implying greater stability in terms of water temperature fluctuations. Such phenomona are intricate, and continuous and long-term observations are essential for a comprehensive understanding of the complex thermal dynamics in estuarine and coastal areas.

## 4. Conclusions

A novel smart temperature sensor (TS-V1) has been developed to improve the accuracy, spatial resolution, and long-term monitoring of estuarine and coastal water temperature. Our sensor integrates a high-precision, low-power CMOS circuit and an integrated chip, resulting in impressive performance during temperature monitoring. Additionally, the sensor has a compact design which ensures its easy production and cost-effectiveness, making it highly practical for widespread deployment. 

The performance evaluation of the TS-V1 sensor showed its good accuracy, high sensitivity, and stability in measuring both water and air temperatures. Specifically, the system exhibited an accuracy rate of ±0.08 °C for water temperature measurements (CTD-Diver had an accuracy of ±0.1 °C), with a sensitivity rate of 0.91 s/°C (with CTD-Diver’s sensitivity values ranging from 0.25 to 0.7 s/°C) and high stability (with a consistency rate of 0.98 for the commercial CTD-Diver sensor). These findings emphasize the reliability and precision of the TS-V1 sensor in accurately measuring water and air temperatures. Further investigations could delve into the underlying mechanisms and technological features that contribute to its superior performance, allowing for its effective utilization in various environmental monitoring applications.

The results of long-term and spatially refined monitoring applications revealed inconsistencies in the distribution of the water and air temperatures in the MDM, JTM, and HTM estuaries during 12 July 2022 to 25 September 2022. A comprehensive range of water temperatures was observed, spanning from 26 °C to 35 °C. Concurrently, the corresponding air temperature variations exhibited a range of 24 °C to 40 °C. The water temperatures at most stations across the three estuaries followed a near-normal distribution, but some are right-skewed. The frequency peak of water temperature is 30~31 °C in the MDM, 29~30 °C in the JTM, and 30~31 °C in the HTM, while peak air temperature is 28~30 °C in the MDM, 29 °C in the JTM, and 28~29 °C in the HTM. Within the Xijiang Estuary area, the middle section of the waterway at the JTM estuary consistently displayed the largest (17.8 °C) differences between water and air temperatures. On the other hand, the outlet of the JTM estuary exhibited the smallest (5.96 °C) disparities between water and air temperatures.

Independently developed, low-cost, and high-precision temperature sensors convincingly address the challenge of limited ambient temperature data in marine environments. The proposed TS-V1 sensor optimizes the CMOS-based circuit chip structure, supplemented by its LDO configuration, enabling high accuracy, low power consumption, and minimal noise. Notably, the sensor displays limited data drift across temperatures, with considerable drift emerging only above 100 °C, rendering it suitable for extended marine environment monitoring. In addition, the titanium alloy shell ensures efficient thermal conductivity and sustained sensitivity. Further bolstered by features such as its cost-effectiveness and streamlined manufacturing process, TS-V1 is well-suited for comprehensive temperature monitoring on a broader scale than the commercial alternatives. In general, this sensor significantly overcomes the fundamental issue of the sparse global temperature data distribution. Additionally, we have developed shore-based fixed temperature sensors and drifting buoy temperature sensors with online real-time visualization capability. These enhancements improve the data acquisition efficiency, safety, and real-time monitoring capabilities in three-dimensional marine, terrestrial, and aerial spaces. More importantly, these advancements have significant implications for marine ecosystem studies and decision-making in marine industries. They also contribute to safer and more comprehensive three-dimensional monitoring across different environments.

## Figures and Tables

**Figure 1 sensors-23-07659-f001:**
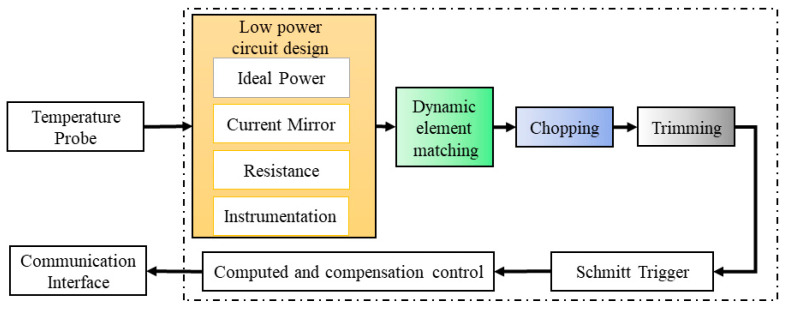
Schematic diagram of leading techniques for low power consumption and the high-precision design of CMOS integrated circuits.

**Figure 2 sensors-23-07659-f002:**
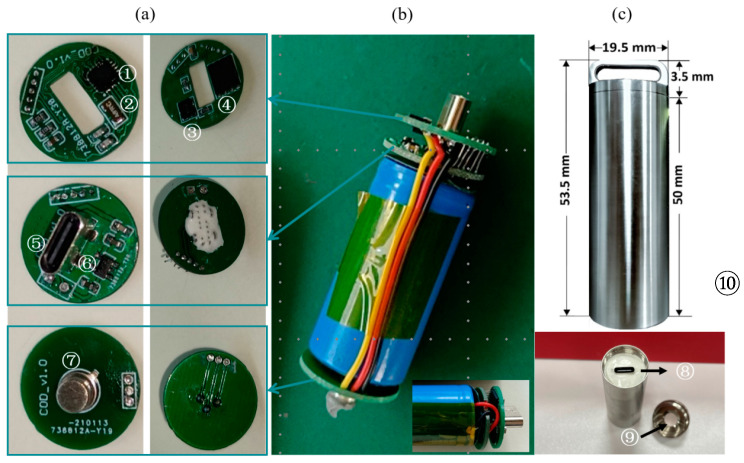
The appearance (**a**), internal structure (**b**), and integrated circuit board (**c**) of TS-V1: low-power Cortex M0 MCU ①; low-speed crystal oscillator ②; high-speed crystal oscillator ③; data storage chip ④; low-power linear voltage regulator chip ⑤; type-c data communication interface ⑥; rechargeable storage battery ⑦; TS-V1 temperature probe ⑧; ritanium alloy shell cap ⑨; titanium alloy shell bottle body ⑩.

**Figure 3 sensors-23-07659-f003:**
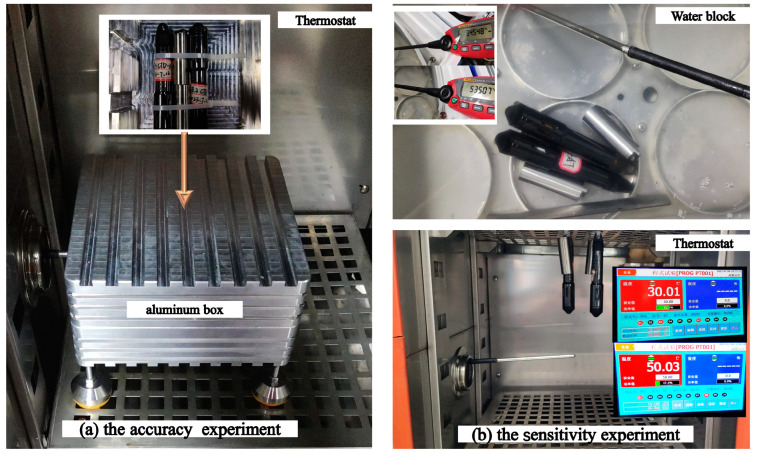
Schematic diagram of accuracy (**a**) and sensitivity (**b**) experiments. TS-V1's titanium alloy shell: thermal conductivity = 7.6 W/(m °C), specific heat capacity = 0.5 kJ/kg °C. CTD-Diver’s zirconia shell: thermal conductivity = 3 W/(m °C), specific heat capacity = 0.8 kJ/(kg °C).

**Figure 4 sensors-23-07659-f004:**
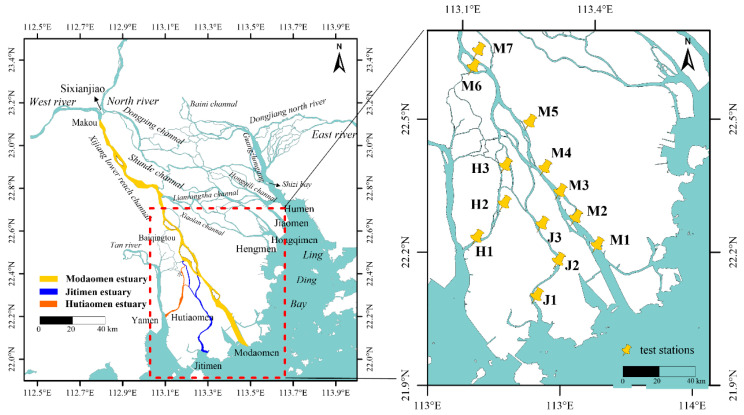
Locations of TS-V1 sensors deployed in the Xijiang Estuary located in the Pearl River Delta. LocationsM1~M7 were deployed in the Modaomen Estuary (MDM), J1~J3 were deployed in the Jitimen Estuary (JTM), and H1~H3 were deployed in the Hutiaomen Estuary (HTM).

**Figure 5 sensors-23-07659-f005:**
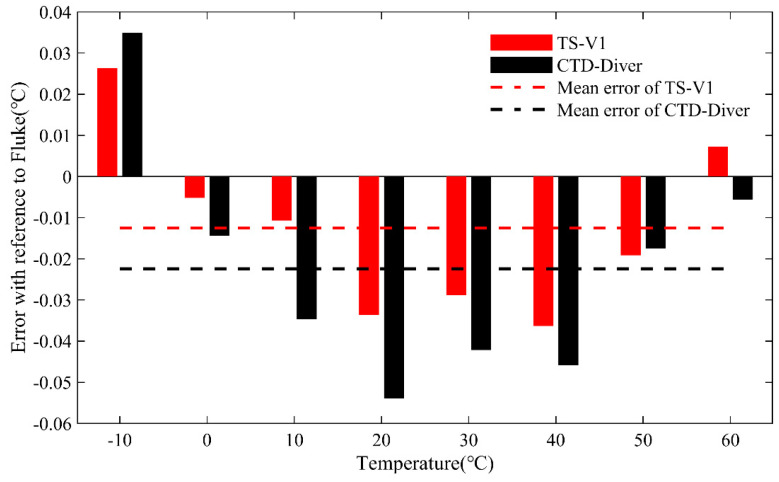
Test results of three different types of temperature sensors on the accuracy over the −10~60 °C range of temperature scenarios.

**Figure 6 sensors-23-07659-f006:**
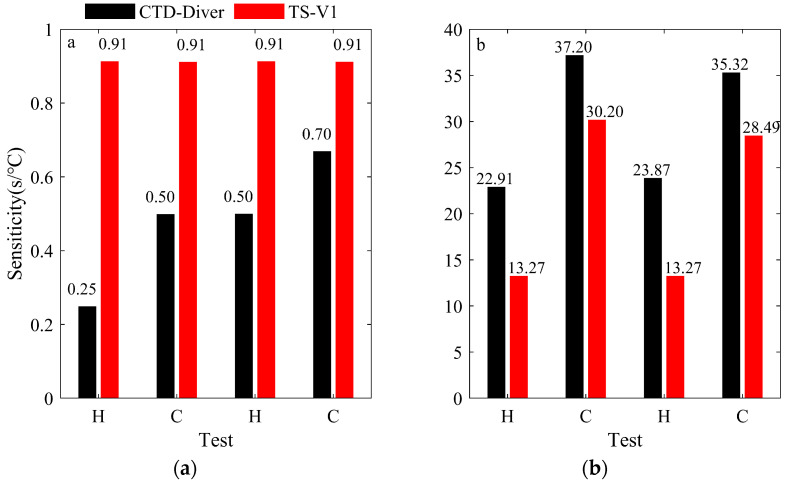
Test results for two different types of temperature sensors regarding their sensitivity to water temperatures (**a**) and air temperatures (**b**). Here, ‘H’ and ‘C’ indicate the heating test and cooling test, respectively.

**Figure 7 sensors-23-07659-f007:**
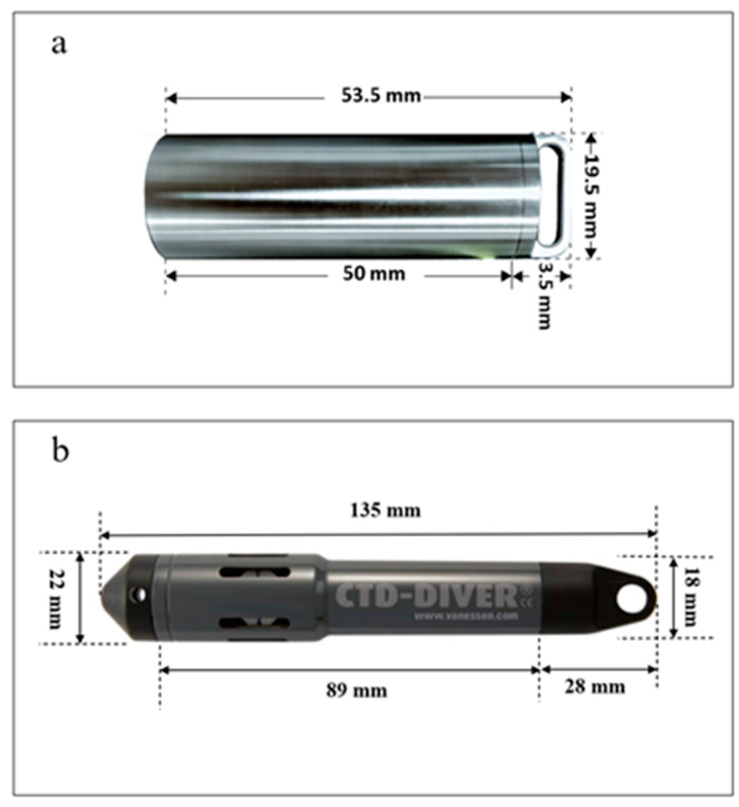
The structures of the TS-V1 (**a**) and CTD-Diver (**b**) sensors.

**Figure 8 sensors-23-07659-f008:**
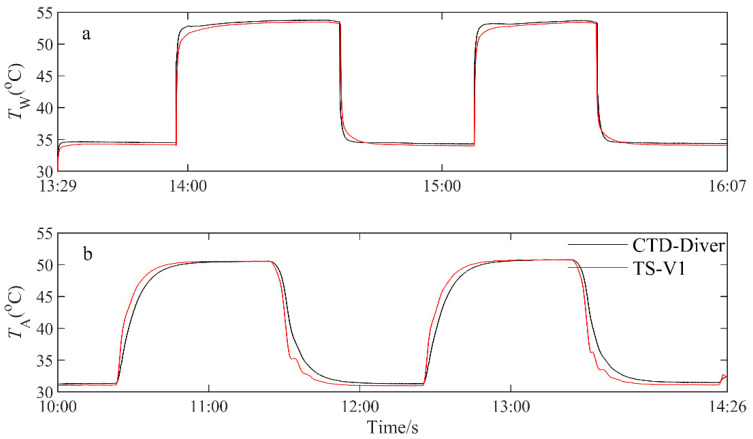
Variations of the water (**a**) and air (**b**) temperatures during the sensitivity test using the TS-V1 and CTD-Diver sensors.

**Figure 9 sensors-23-07659-f009:**
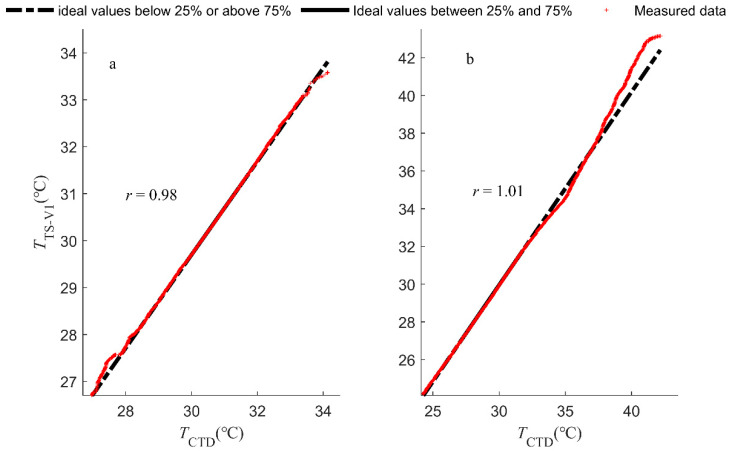
The QQ plots of the water (**a**) and air (**b**) stability assessments comparing the observations between CTD-Diver and TS-V1.

**Figure 10 sensors-23-07659-f010:**
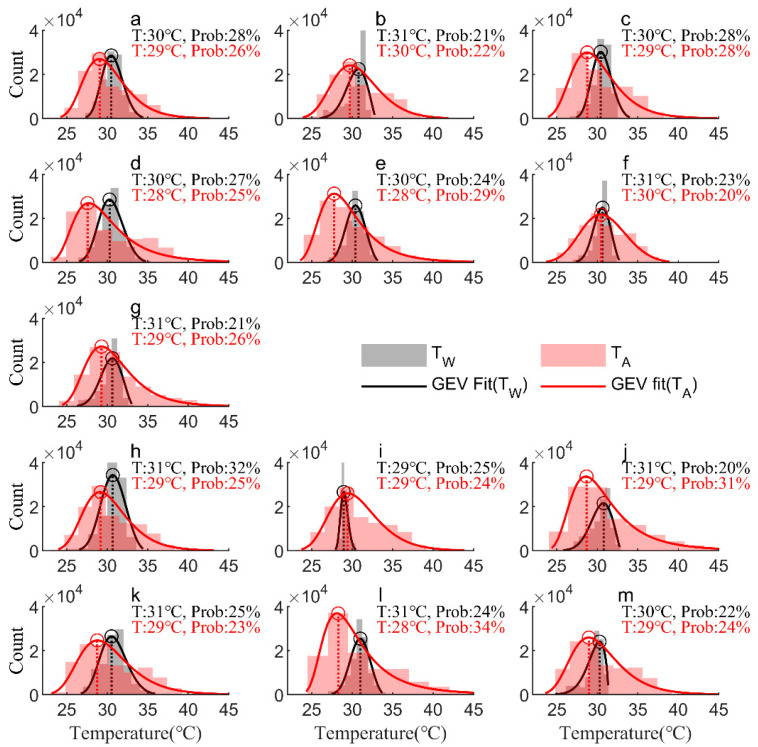
The probability density functions (PDFs) of the GEV distribution corresponding to the observed water temperature (TW) and air temperature (TA) values at 13 stations: (**a**–**g**) M1~M7 in the MDM; (**h**–**j**) J1~J3 in the JTM; (**k**–**m**) H1~H3 in the HTM (here, the black and red circles on the diagram indicate the highest probability temperatures during the study period, with corresponding water and air temperatures provided separately).

**Figure 11 sensors-23-07659-f011:**
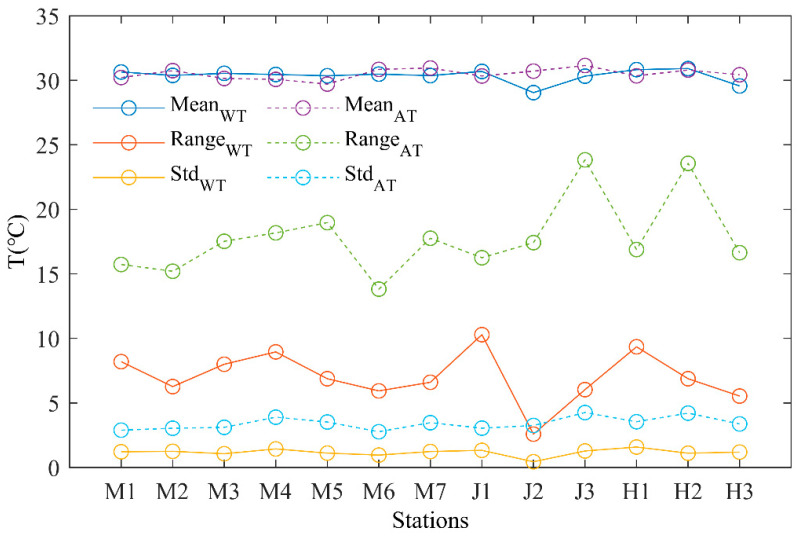
The mean, range, and standard deviation (Std) values obtained from observations in the Xijiang channel networks.

**Table 1 sensors-23-07659-t001:** The key technological indicators of the chips.

Parameters	Technological Indicators
High accuracy	0.1 °C (−20 °C to 80 °C)
Extremely low noise	less than 0.001 °C
Wide temperature range	−45 °C to 130 °C (accuracy error maximum is 0.3 °C)
Wide supply voltage range	2.7 V to 5.5 V
Ultra-low current	60 µA active or 220 nA average

**Table 2 sensors-23-07659-t002:** Calibrated parameters of the GEV distribution.

Parameters	M1	M2	M3	M4	M5	M6	M7	J1	J2	J3	H1	H2	H3
*g* _WT_	−0.22	−0.47	−0.23	−0.23	−0.32	−0.38	−0.41	−0.30	−0.19	−0.48	−0.19	−0.33	−0.63
*g* _AT_	−0.04	−0.11	−0.01	0.13	0.09	−0.22	0.03	−0.05	−0.07	0.16	−0.02	0.21	−0.02
*m* _WT_	30.18	30.04	30.12	29.91	29.97	30.17	30.00	30.20	28.87	29.99	30.21	30.54	29.35
*m* _AT_	28.91	29.45	28.69	28.06	27.99	29.79	29.34	28.97	29.27	28.99	28.71	28.59	28.87
*s* _WT_	1.22	1.33	1.15	1.43	1.17	1.04	1.32	1.45	0.42	1.37	1.53	1.17	1.26
*s* _AT_	2.38	2.65	2.51	2.81	2.54	2.61	2.66	2.54	2.74	2.86	2.90	2.73	2.75

## Data Availability

The data supporting the findings of this study are available from the first author or corresponding author upon reasonable request.

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
