# Peer review of "Smart Temperature Sensor Design and High-Density Water Temperature Monitoring in Estuarine and Coastal Areas"

_sensors, 2023, doi:10.3390/s23177659_

Round 1

Reviewer 1 Report

The authors demonstrate smart temperature sensor for water temperature monitoring in esturrine and coastal areas. The paper has some interest for the field of real application of temperature sensing. There are some questions before the consideration of publishing in this Sensors journal.

1) The authors do not introduce the detailed structure of sensing component and its sensong principle.

2) What’s novelty of this proposed temperature sensor relative to those commerical products, although the authors compare it with the other two products. Some more explanation could be added in the manuscript.

3) Ther are no marked instructions in Fig. 9.

4) I suggest adding at the Authors’ reference of the recent reference about high-sensitivity temperature sensor of “IEEE Sensors Journal, 23(5), 4843-4848, 2023”.

5) The authors could give some results of the relationships between the voltage variations of proposed sensors and the temperature perturbation in the manuscript.

Author Response

Dear reviewer,

We thank you for the careful consideration of our work. Your constructive and thoughtful comments and suggestions led to a much improved and complete revision of the manuscript. In the revised paper, we have addressed all the comments formulated by the Reviewer by replying (in black) to the remarks (in blue). The line numbers in this rebuttal refer to the revised version of the manuscript. Please find our detailed responses to all the comments raised by the reviewer in the attached PDF file.

Thank you very much for your time and consideration.

Yours sincerely,

Huayang Cai (on behalf of all co-authors) 

Reviewer 2 Report

Dear Authors,

Thank you for such important research in the field of semiconductor temperature sensors. You have presented very necessary data in practice. You've done quite a lot of work. But I have a lot of questions and comments about the presentation of your data in this manuscript. Separately, it is worth paying attention to the proof of the relevance of using this temperature sensor chip. Also, please pay more attention to measurement errors between groups of sensors under different conditions. I left all my comments in the attached file "sensors-2483829-peer-review-v1 (Review)".

I hope for your understanding and wish you success in your future research. I will be waiting for your prompt reply point-to-point.

Best wishes,

Reviewer

Author Response

Dear reviewer,

We thank you for the careful consideration of our work. Your constructive and thoughtful comments and suggestions led to a much improved and complete revision of the manuscript. In the revised paper, we have addressed all the comments formulated by the Reviewers by replying (in black) to the remarks (in blue). The line numbers in this rebuttal refer to the revised version of the manuscript. Please find our detailed responses to all the comments raised by the reviewer in the attached PDF file.

Thank you very much for your time and consideration.

Yours sincerely,

Huayang Cai (on behalf of all co-authors) 

Round 2

Reviewer 2 Report

Dear Authors,

Thank you for answers. I have got all necessary points from you. I wish you good luck in your future researches.

Kind regards,

Reviewer